https://doi.org/10.1038/s42003-020-01132-8　　　**OPEN**

# Detecting protein and post-translational modifications in single cells with iDentification and qUantification sEparaTion (DUET)

Yandong Zhang[1,2], Changho Sohn [1,2], Seoyoen Lee[1], Heejeong Ahn[1], Jinyoung Seo [1], Junyue Cao[1] & Long Cai [1✉]

While technologies for measuring transcriptomes in single cells have matured, methods for measuring proteins and their post-translational modification (PTM) states in single cells are still being actively developed. Unlike nucleic acids, proteins cannot be amplified, making detection of minute quantities from single cells difficult. Here, we develop a strategy to detect targeted protein and its PTM isoforms in single cells. We barcode the proteins from single cells by tagging them with oligonucleotides, pool barcoded cells together, run bulk gel electrophoresis to separate protein and its PTM isoform and quantify their abundances by sequencing the oligonucleotides associated with each protein species. We used this strategy, iDentification and qUantification sEparaTion (DUET), to measure histone protein H2B and its monoubiquitination isoform, H2Bub, in single yeast cells. Our results revealed the heterogeneities of H2B ubiquitination levels in single cells from different cell-cycle stages, which is obscured in ensemble measurements.

[1] Division of Biology and Biological Engineering, California Institute of Technology, Pasadena, CA, USA. [2] These authors contributed equally: Yandong Zhang, Changho Sohn. ✉email: lcai@caltech.edu

Protein and its post-translational modification (PTM) states are often involved in dynamic and oscillatory processes that are not synchronized in a population and show substantial single-cell heterogeneities[1–4]. Conventional protein-detection methods, such as western blots, enzyme-linked immunosorbent assay (ELISA) are generally difficult to downscale to the single-cell level to detect such heterogeneities[5]. Antibody-based methods rely on the availability of good affinity reagents which are often the limiting factor in experiments[6]. Recently, great advances have been made in single-cell mass spectrometry analysis using reporter ions and cell pooling[7] with potential for further increases in sensitivity, throughput, and coverage.

Here, we reported an alternative approach to accurately detect protein and its PTM isoforms in single cells. We reasoned that if the identification and quantification of single-cell protein analysis could be separated into two distinct steps, we could overcome the limitations of conventional biochemical techniques. Specifically, after barcoding proteins from single cells with oligonucleotides, we pool a large number of cells together to yield enough material. Then different protein species, such as target protein and its PTM could be identified by conventional biochemical analysis (e.g. western blot). Despite pooling, the single-cell identities of each protein molecule are preserved in the barcodes. The barcodes from each protein species can be quantified by next-generation sequencing (NGS) to determine the abundance of those proteins in single cells (Fig. 1a). Therefore, iDentification/qUantification SeparaTion (DUET) could combine the powers of conventional biochemical analysis methods for protein identification and next-generation sequencing for protein quantification. As a proof-of-concept experiment, we used this strategy to measure histone protein (H2B) and its monoubiquitination isoform (H2BK123ub) in single yeast cells.

## Results

**In situ protein-oligonucleotide ligation**. To perform DUET, we need to tag and barcode proteins from single cells to preserve their cellular identities throughout the entire process. We first tag a DNA oligonucleotide to histone protein (H2B) in fixed cells using spytag/spycatcher system[8] (Fig. 1b). Spytag is a 13-amino-acid peptide that can form an isopeptide with its complementary protein, Spycatcher, with high efficiency and specificity. To test the in situ DNA oligo tagging, we constructed *S. cerevisiae* yeast strains containing spytag and 3xFLAG at the C terminal of H2B (see "Methods" section). We then synthesized spycatcher-DNA oligonucleotide conjugate using *trans*-cyclooctyne/methyltetrazine click chemistry (Supplementary Fig. 1). The western blot of whole-cell lysate using anti-FLAG antibody shows the H2B protein bands with slower migration after reacting with spycatcher-DNA oligo conjugate, indicating that spycatcher-oligos conjugate can diffuse into fixed cells and conjugate efficiently to the spytag on the protein (Fig. 1d). We tested several other different target proteins with different copy numbers and with different cellular localizations (nucleus: H2B, 101,430 copies per cell[9]; cytoplasm: Pre1, 13,312 copies per cell[9]; Snf1: 5423 and Glc7: 12,111 per cell, respectively[9]) (Fig. 1d), and the in situ tagging efficiencies were high for all the proteins tested with minimal unreacted proteins (Fig. 1d; Supplementary Fig. 2).

**Single-cell barcoding by combinatorial indexing**. Second, we use a combinatorial indexing scheme[10] to uniquely barcode the proteins from single cells. The combinatorial indexing allows a large number of cells to be barcoded uniquely and avoids manipulation of individual cells as the experiments could be performed with normal pipetting. In the experiments, two sequential rounds of "split-pool" T7 ligation were performed to ligate barcoded oligo adaptors to the proteins inside the cells (Fig. 1b, c). The cells were split into 96 wells each with a distinct oligo that was ligated to the spycatcher-oligo via a ligation bridge (Fig. 1c; Supplementary Data 1). Cells were then pooled and split again into another 96 wells and ligated with another set of oligos. After two rounds of barcode ligations, $96^2 = 9216$ barcodes are possible. To ensure we sample cells that have unique barcodes, we aliquoted 900 cells to decrease barcode collisions rate (<5%, Supplementary Data 2). Western blot shows that the targeted protein bands shift up sequentially after two rounds of ligation, indicating that the barcode oligo is successfully ligated (Fig. 1d). Similar with the spytag-spycatcher reaction, the T7 ligation efficiency is also high for all the proteins tested with little initial un-ligated protein visible on the gel (Fig. 1d; Supplementary Fig. 2). In addition to the cellular barcodes, we also incorporated a 12 nt random-base Unique Molecular Identifier (UMI)[11] sequence in the first round of DNA barcode oligo (Fig. 1c), which will be used to correct the amplification biases in NGS library generation. Finally, we validated that the cell morphology was well preserved after in situ oligo tagging and two rounds of ligation (Supplementary Fig. 3).

**Identification of protein isoforms by conventional biochemical separation**. Third, to generate enough proteins for conventional protein separation methods, such as SDS-PAGE, the barcoded cells were pooled together. Despite pooling, the single-cell identities of these proteins are still preserved in the covalently attached oligo. In the experiment, we performed SDS-PAGE to separate H2B and its monoubiquitinated isoform H2Bub. H2Bub is 7 kDa heavier than H2B, appearing as a heavier band in the gel (Fig. 1d). Then, the H2B and H2Bub bands were excised from the gel and the protein-oligo conjugates were extracted from the gel pieces (Fig. 1a). To boost the gel staining intensity from the small number of barcoded cells (~900 cells) and to reduce nonspecific sample loss, we co-loaded $10^6$ "dummy" cells, in which the H2B proteins were tagged with oligos, which was of the same length as the real barcode but did not contain PCR primers. As a result, the protein-oligo conjugates originated from dummy cells would co-migrate with those from barcoded cells, but would not be amplified by PCR (Supplementary Fig. 4). The dummy oligo also had a TAMRA dye at the 3′ end for band visualization on the gel (Supplementary Fig. 4). To achieve high-efficiency recovery, polyacrylamide gel with a reversible crosslinker was used[12] (see "Methods" section).

**Single-cell protein isoform quantification by sequencing**. Finally, to quantitation of protein isoform abundance in single cell, we sequenced the library of single-cell barcodes generated by PCR of the extracted protein-oligo conjugates. We firstly identified the real-cell barcodes by plotting the total number of sequencing reads per barcode in descending order (Fig. 2a). We observed a clear cutoff to separate real-cell barcodes with a high number of reads from spurious cell barcodes with a low number of reads. The spurious barcodes likely stem from PCR and sequencing errors. We identified 850 real-cell barcodes, which agrees with our experimental design (~900 barcoded cells are aliquoted). The same set of cell barcodes were also identified from the H2Bub gel band (848 out of 850 H2B cell barcodes) (Supplementary Fig. 5), further confirming that those barcodes represent real cells. After identifying barcodes corresponding to the real cells, we next quantified the protein copy numbers in those cells based on the UMIs associated with the cell barcodes. To demonstrate that the copy number of proteins per cell detected is accurately reflected by the number of UMIs[13], we

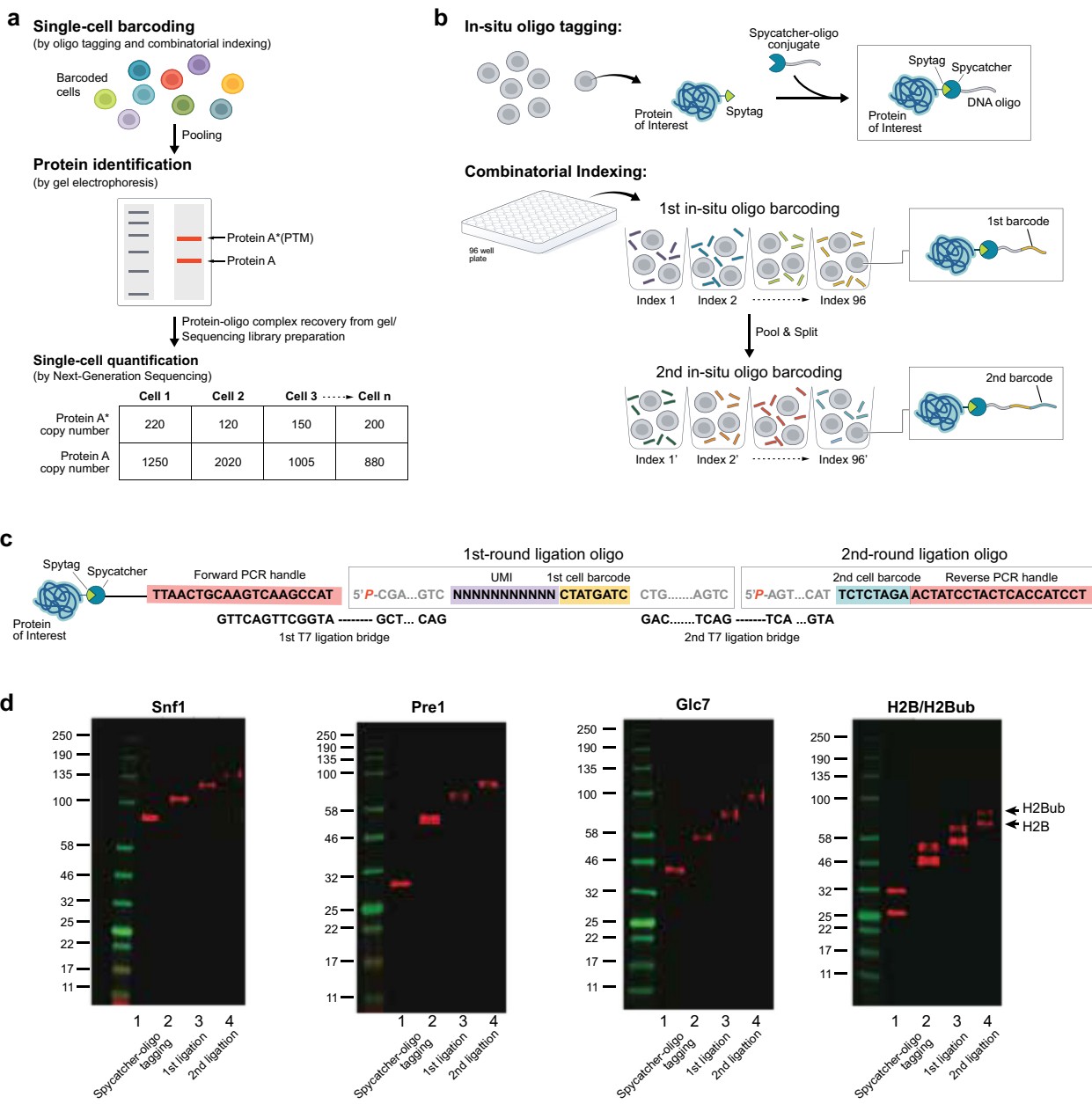

**Fig. 1 Schematics of iDentification/qUantification sEparaTion (DUET). a** The cells are uniquely barcoded by oligo tagging. Then the cells are pooled together, analyzed by gel electrophoresis to identify protein and post-translational modifications. After protein identification, the protein-oligo conjugates are recovered and the oligo part is amplified by PCR to generate next-generation sequencing libraries. The single-cell protein abundance can be quantified from the sequencing results. **b** *S. cerevisiae* strain containing spytag at the C terminal of the targeted protein is constructed and the fixed cells are reacted with spycatcher-oligo to covalently attach DNA oligo to targeted proteins in situ. Then the cells are combinatorially indexed with two rounds of "split-pool" barcoding. The cells are firstly distributed into a 96-well plate, and well-specific barcodes were ligated to the DNA oligo on the proteins via T7 ligation. Then the cells were pooled together and randomly distributed again into another 96-well plate where second barcodes were ligated. **c** The oligo design. The oligo in spycatcher-DNA oligo conjugate is 20 nt, which serves as the PCR forward primer binding site during sequencing library generation. The 5'-phosphorylated 1st ligation barcode oligo contains a ligation site (10 nt, for 1st round ligation), a UMI sequence (12 nt), a cell barcode (8 nt), and another ligation site (17 nt, for 2nd round ligation). The 5'-phosphorylated 2nd ligation barcode oligo contains a ligation site (17 nt, for 2nd round ligation), a cell barcode (8 nt) and the reverse PCR primer binding site (20 nt). The ligation bridge sequences are complementary to ligation sites. **d** Western blot analysis of different target proteins (Snf1, Pre1, Glc7, and H2B) with sequential reactions with spycatcher-oligo, the first ligation and the second ligation, respectively. For H2B protein, H2B (lower band) and its monoubiquitination isoform H2Bub (upper band) are separated as they have different molecular weights.

showed that (1) the length of UMI (12 nt) has enough coding space to cover the protein abundance in single cells (Supplementary Fig. 6a, b); (2) sequencing depth is sufficient to sample all the possible UMIs (Supplementary Fig. 6a–c). As a result, UMIs efficiently eliminate the PCR amplification bias and ensure accurate quantification. We detected an average of $4568 \pm 2004$ (s.d.) UMIs per cell using 10% of the extracted materials (see "Methods" section; Supplementary Data 3). There are estimated $101,430 \pm 63,961$ copies of H2B per cell[9]. From this, we estimate that the detection efficiency of DUET to be $45 \pm 20\%$.

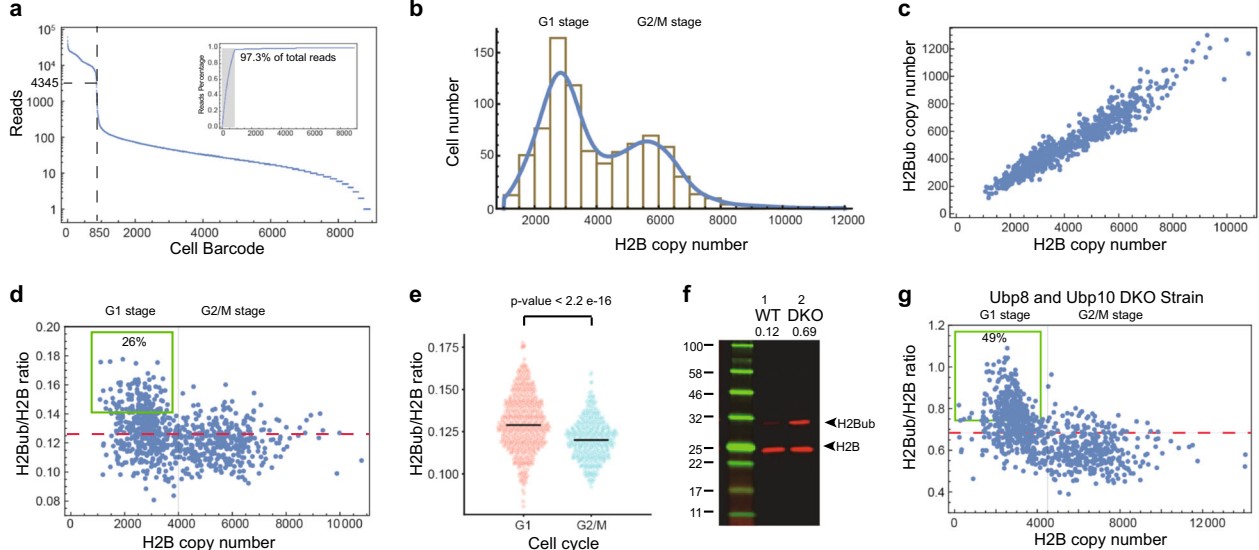

**Fig. 2 Quantification of H2B and its monoubiquitination H2Bub abundances in single yeast cells. a** Cell barcode identification from sequencing results. The number of reads per cell barcode was plotted in the descending order. A clear cutoff (dashed line) could be identified to separate real barcodes (with a high number of reads) from the spurious cell barcodes (with a low number of reads). The inset shows accumulated reads percentage. The gray area corresponds to real-cell barcodes, which account for 97.3% of the total filtered reads. **b** The histogram of H2B copy numbers in single cells. **c** H2B and H2Bub copy number in single cells. Each dot represents a single cell. **d** The ratio of H2Bub/H2B in single cells. The red dashed line indicates the population-average H2Bub/H2B ratio. The gray line divides the cells into G1 and G2/M cell-cycle stages according to the H2B copy number. The green box indicates the more-ubiquitinated populations in the G1 stage. **e** The distributions of H2Bub/H2B ratio for cells in G1 and G2/M stages, respectively. The two stages have different distributions (Welch's $t$-test, $n_1 = 365$, $n_2 = 484$). **f** Western blot images of H2B for the wild-type strain (WT) and the UBP8 and UBP10 double-knockout strain (DKO). The H2Bub/H2B ratios measured from the image are ~12% in WT and ~69% in DKO, respectively. **g** The H2Bub/H2B ratios as a function of H2B copy number in single cells for DKO strain. Each dot represents a single cell. The red dashed line is the population-average H2Bub/H2B ratio. The green box indicates hyper-ubiquitinated populations in the G1 stage.

**Single-cell H2B ubiquitination heterogeneities**. Our method allows us to quantify the heterogeneity in the copy numbers of H2B (Fig. 2b, c). Previous literature reported that H2B copy number is doubled in G2/M cells compared to G1 cells[14,15]. The histogram of the H2B copy number measured from our method shows a bimodal distribution. The two populations have approximately 2-fold difference in the average H2B copy numbers. Therefore, we deduced that the two populations correspond to cells being either in G1 or G2/M cell-cycle stages. This result indicates that our method could accurately quantify the copy number of H2B in single cells (Fig. 2b).

DUET also allows us to quantify the monoubiquitination isoform (H2Bub) within the same single cells. We calculated the ratio between H2Bub to H2B in single cells (Fig. 2d). Interestingly, cells at different cell-cycle stages show different H2Bub/H2B ratio distributions (Fig. 2d, e). Specifically, 26% of cells at the G1 stage have high ratio of H2Bub to H2B (>0.14), compared with only 5% of cells at the G2/M stages. To further investigate the molecular mechanism underlying H2Bub/H2B ratio change in G1 and G2/M cells, we applied our method to a double-knockout yeast strain in which two de-ubiquitination enzymes of H2B (Ubp8 and Ubp10)[16] are deleted. As expected, the H2Bub level was elevated in this strain (H2Bub/H2B ratio: 0.69), estimated from ensemble western blot (Fig. 2f). However, 49% of cells at the G1 stage still have the high ratio of H2Bub to H2B (>0.74), compared with 5% of cells at the G2/M stage (Fig. 2g). These results suggest that Ubp8 and Ubp10 set the baseline levels of ubiquitination of H2B, whereas the cell-cycle dynamics of H2B ubiquitination may be due to other components. It has been shown that Rad6, the E2 ubiquitin ligase for H2B, can be phosphorylated by Bur1/Bur2 cyclin-dependent protein kinase at its serine-120 residue[17]. Therefore, the role of Rad6 and Bre1, the E3 ubiquitin ligase for H2B, in modulating levels of H2Bub should be further investigated.

## Discussion

In summary, this proof-of-concept experiment demonstrates that DUET strategy could detect protein and its PTM isoforms in single cells. The DUET approach does not require manipulation of single cells and can be performed with common lab equipment. DUET has high detection efficiency (~45%) where most of the loss presumably occurred during sample handling and gel electrophoresis. Such high detection efficiency will in principle allow low-copy number protein to be detected in single cells. In DUET, alternative methods of protein separation and identification, e.g., phos-tag gel[18], can be explored so that other post-translational modification isoforms, such as phosphorylation, can be resolved and quantified at single-cell level. For this initial demonstration of DUET, we used genetic fusion to selectively target a single protein. In the future, DUET could be extended to label the proteome globally by methods such as incorporation of unnatural amino acids[19]. The unnatural amino acids could be clicked to oligonucleotide and barcoded using the same strategy demonstrated here. Then, different proteins-oligo conjugates could be separated by high dimensional gels[20] or liquid chromatography[21], followed by identification by mass spectrometry and quantification by NGS of the individual isolated protein bands, enabling single-cell proteomics at a global scale.

## Methods

**Yeast strains and plasmids**. The *S. cerevisiae* strains used in this study were BY4741 (MATa *his3 leu2 met15 ura3*). The standard cloning procedure was performed[1] to tag the C terminal of target protein with spytag and 3xFLAG. The strains and plasmids are available upon request.

**Cell culture, fixation, and permeabilization**. Fresh colonies of yeast strain were grown in YPD until OD600 of ~0.5 (10 mL culture). Cells were then fixed by 1% w/v formaldehyde (Thermo Scientific, 28908) at 30 °C for 30 min with gentle shaking. Cells were then harvested and washed by buffer B (1.2 M sorbitol/0.1 M

sodium phosphate, pH 7.4) three times. The cells were spheroplasted using 100 μg zymolase (Zymo Research, E1006) and 10 μL fresh beta-mercaptoethanol in 1 mL of buffer B cell suspension for 10 min at 37 °C with gentle shaking. After the spheroplasting reaction, the cells were gently washed with buffer B three times. Cells were post-fixed in 1% w/v formaldehyde in 1X PBS/0.6 M KCl for 30 min at RT. Cells were washed with buffer B three times again after post-fixation.

**Spycatcher-DNA oligo conjugate synthesis**. The strategy for synthesizing spycatcher-DNA oligo conjugate is shown in Supplementary Fig. 1. Spycatcher with 6xHis-tag and a cysteine sequence at C terminal was expressed in the derived BL21 strain (NEB, C2566H, T7 express competent *E. coli*) and purified using standard Ni-NTA purification method. To prepare spycatcher-methyltetrazine, spycatcher was reduced by TECP (Thermo Scientific 77720) to remove the potential intermolecular disulfide bond. Excessed TCEP was then removed by PD-10 desalting column (GE Healthcare). The spycatcher was reacted with maleimide-(PEG)$_4$-methyltetrazine (Click Chemistry Tools, 1068-10) via a free thiol group in the reduced cysteine residue and the reaction product (spycatcher-methyltetrazine) was separated from unreacted maleimide-(PEG)$_4$-methyltetrazine by PD-10 column. To prepare *trans*-cyclooctyne (TCO)-oligo, 5′-amine-modified oligonucleotide (IDT DNA) was reacted with TCO-(PEG)$_4$-NHS ester (Click Chemistry Tools, A137-2) and the reaction mixture was purified by HPLC using a C8 column. Finally, to prepare spycatcher-DNA oligo conjugate, spycatcher-methyltetrazine was reacted with eq. molar amount of TCO-oligo via the click chemistry between methyltetrazine and TCO (Supplementary Fig. 1a). Spycatcher-oligo conjugate was purified from unreacted spycatcher and TCO-oligo by ion-exchange chromatography (Supplementary Fig. 1b) and stored with 50% glycerol in 1X PBS at −20 °C until further usage.

**In situ DNA oligo tagging**. 10 μM spycatcher-DNA oligo conjugate was reacted with the fixed cells in 1X PBS/0.6 M KCl solution containing protease inhibitor cocktail (Sigma Aldrich, SRE0055). The formaldehyde fixation and spheraplasting process permeabilized the cells so that spycatcher-oligo conjugate could enter the cells and react with protein with spytag. The reaction was incubated for 2 h at RT with gentle shaking. After the spycatcher-oligo reaction, the cells were washed with buffer B three times.

**Pool-split combinatorial barcoding with T7 ligation**. Cells after in situ DNA tagging were distributed into a 96-well plate. Each well contains ~10⁶ cells. T7 ligation reaction buffer containing T7 ligase (NEB, M0318S), 1st round ligation adapter (5 μM) and 1st round barcoding oligos (5 μM) were added into each well. The plate was incubated for 2 h at RT with gentle shaking. After 1st barcode ligation, cells were pooled together, washed with buffer B three times, and distributed into another 96-well plate. T7 ligation reaction buffer containing T7 ligase, 2nd round ligation adapter (5 μM) and 2nd round barcoding oligos (5 μM) were added into each well. The plate was incubated for 2 h at RT with gentle shaking. All barcode sequences used in this work were acquired from NEB-Next 96 single index kit barcode sequences (NEB, E6609), listed in Supplementary Data 2. After 2nd round barcode ligation, cells were pooled together and washed with buffer B three times. The cell morphology was checked under the microscope after spycatcher-oligo conjugation, 1st ligation, and 2nd ligation to make sure the cells remain intact during this procedure (Supplementary Fig. 3). The cell density was measured using a hemocytometer and a cell-suspension solution containing 900 cells was aliquoted using flow cytometry.

For "dummy" sample preparation, we first synthesized spycatcher-DNA oligo conjugate with the dummy sequence using the same method as described previously. Then cells were reacted with the spycatcher-dummy oligo, ligated sequentially with 1st round barcode oligos and 2nd round barcode oligos using the same methods as before, but without pool-splitting. The dummy sample has different sequence in the PCR handle parts so that it will not be amplified by primers for Illumina sequencing library preparation (Supplementary Fig. 4a). In addition, the 3′ end of 2nd ligation oligo is modified with a rhodamine dye TAMRA, to enable visualization of the ligation bands in gel analysis by a typhoon scanner (Supplementary Fig. 4b, c). The dummy sample was mixed with the aliquot of real barcoded sample (~900 cells) for further analysis.

**Gel electrophoresis and protein–DNA complex recovery**. 2X Laemmli buffer (Bio-Rad, 1610737) was added to the cells (containing both dummy cells and barcoded cells) and boiled at 95 °C for 10 min. This boiling process reversed the formaldehyde crosslinking. The sample was then loaded in a 10% dissolvable polyacrylamide gel. The dissolvable PAGE gel was made with a labile crosslinker, ethylene-glycol-diacrylate (EDA) (Sigma Aldrich, 41608), which allows for high recovery yield from the gel[2]. The target protein-oligo conjugate bands were visualized using a Typhoon scanner to image with TAMRA fluorescence. The bands were cut off from the gel, and the protein–oligo complex were recovered. We also cut and extracted a blank gel piece (Supplementary Fig. 5a) to estimate the background introduced during gel electrophoresis.

**Library preparation and sequencing**. Two rounds of PCR amplification were carried out for next-generation sequencing library preparation. 10% of the materials

recovered from the gel was used for PCR amplification. First, the DNA part of the protein–DNA conjugate was amplified using its PCR handle. Then in second-round PCR, sequencing adapters were appended using NEB-Next Multiplex Oligos for Illumina (NEB). The amplification conditions for the first-round PCR were as follows: 95 °C 1 min, then 10–15 cycles at 95 °C, 10 s/62 °C, 15 s/65 °C 30 s, and a final extension at 65 °C 3 min. The number of cycles required for the first-round PCR was determined by analyzing a small aliquot of the sample on a qPCR machine. The number of cycles was determined as the start point of exponential phase amplification. The PCR amplification condition for the second-round PCR was as follows: 95 °C 1 min, then 4 cycles at 95 °C 10 s, 62 °C 15 s, 65 °C 30 s, and a final extension at 65 °C 3 min. After each round of PCR, PCR amplicons were separated on 3% agarose gel and purified using gel extraction kit (Thermo Scientific, K210012) without heating. The PCR-amplified library was quantified using a Qubit High-sensitivity DNA kit (Invitrogen). The final purified amplicons were sequenced using a HiSeq 2500 (Illumina) with the targeted read depth of 5–25 million per gel band.

**Data analysis**. To estimate the "collision" rate (the number of barcodes representing more than two cells), we simulated the sampling process (Supplementary Data 2) using the procedure described in the previous work[3]. We found that with 9216 possible barcode combinations, the sampling of 900 cells will result in an expected collision rate lower than 5%. Therefore, we aliquoted 900 cells in the experiment for the following analysis.

The sequencing reads were first filtered based on the constant fixed region in the oligo (the constant region includes the PCR handle, the first T7 ligation site, and second T7 ligation site). Reads that had more than one mismatch against the constant region were disregarded. Then, the 1st round cell barcode and 2nd round cell barcode were connected together to generate the full cell barcode. Reads with cell barcodes which did not match the set of barcode combinations (9216 in total) were disregarded. The number of reads for each barcode was then calculated and the real-cell barcodes were identified from spurious cell barcodes as the former have a much higher number of reads than the latter (Supplementary Fig. 5b). Although the real barcodes could be found from both H2B sample and H2Bub sample, they cannot be found from the background sample (Supplementary Fig. 5c). In addition, the number of unique UMIs is significantly lower in the background band compared with the targeted protein band, indicating the gel background is low.

To verify that the UMIs had enough coding space to encode all the proteins in single cells, we counted how many unique UMIs we could identify from sequencing results when we computationally shortened the UMIs (Supplementary Fig. 6a, b). The number of UMIs increased with the length of the UMIs and reached a plateau after around 10 nt, indicating that the length of UMI (12 nt) have enough coding space to encode all proteins in single cells. To verify that the sequencing depth was high enough to sample all the UMIs, we computationally subsampled the sequencing reads and calculated how many UMIs observed were associated with single-cell barcodes (Supplementary Fig. 6c). As sequencing depth increases, the number of uniquely identified UMIs increases and reached a plateau at full sequencing depth (1.0), indicating that all the UMIs are sufficiently sampled. It should be noted that different sequencing depths were needed for different proteins to saturate the UMIs. For example, for the H2B sample, 25 million reads were needed, while for the H2Bub sample, only 5 million reads were required for library saturation. This reflects the different complexity of these two libraries, which agrees with the different copy numbers of these two proteins inside the cells.

**Statistics and reproducibility**. ImageJ software was used to analyze western blot images. Single-cell protein copy number data were processed using Microsoft Excel. Results were shown as mean ± S.D. All statistical analyses were depicted in the figure legends. Statistical significance was assessed using Welch's *t*-test. *p*-values of 0.05 or less were considered statistically significance and absolute *p*-value is presented in figures.

**Reporting summary**. Further information on research design is available in the Nature Research Reporting Summary linked to this article.

## Data availability

Next-generation sequencing data for H2B and H2Bub from wild-type and double-knockout strains are available at the NCBI's GEO (Accession Number: GSE153605). The single-cell protein copy number counts used in our analysis are available in Supplementary Data 3. All other relevant data are available from the corresponding author upon request.

## Code availability

All software and codes used in this work will be made available upon request to the authors.

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

## Acknowledgements

We thank R. Ismagilov for helpful suggestions and discussions. We thank I. Anto-schechkin from the Millard and Muriel Jacobs Genetics and Genomics Laboratory for assistance with the sequencing. We thank I. Strazhnik for help preparing the figures. This work was supported by funding from Paul G. Allen Frontiers Foundation Distinguished Investigator and Discovery Center.

## Author contributions

L.C. conceptualized the project. Y.Z., C.S., and S.L. performed the laboratory experiments. H.A., J.S., and J.C. helped prepare the reagents and materials. Y.Z. did the analysis and visualized the data. Y.Z. and L.C. wrote the draft.

## Competing interests

The authors declare no competing interests.
