## [Peer Review File · Communications Biology]

Reviewers' comments:

Reviewer #1 (Remarks to the Author):

Zhang and co-authors develop a method to analyze post-translational modified populations of a protein at the level of single cells. It combines the use of DNA bar codes and pooling to protein gel electrophoresis and NGS for DNA sequencing. They applied the method to detect histone 2B and its ubiquitinated form. Indeed, a method to analyze PTM of proteins in single cells is lacking and would be of great interest to the community. However, the manuscript needs clarifications prior to publication.

The manuscript is difficult to follow.

It would help the reader if the rationale of each step is explained. Even scheme B of Figure 1 is not clear and it is the main figure.

Why two rounds of ligation are done?

It's not clear how it is determined which cells are in G1 and which in G2M. How copy number is measured.

How this analysis is related to single cell information given the accuracy (Authors detected on average of 4065 ± 1798 UMI of H2B proteins per cell with a detection efficiency to be $40 \pm 18\%$)

Finally, the methodology is limited to proteins that are tagged so it cannot be applied to native proteins. This is a big limitation to its application.

Reviewer #2 (Remarks to the Author):

In this manuscript entitled "Detecting protein and post-translational modifications in single cells with iDentification and qUantification sEparaTion (DUET)", Yandong Zhang and colleagues describe a new method they coin "DUET" that can be used to detect protein and its PTM isoforms in single cells. The authors innovatively labeled protein with oligonucleotides and combined NGS sequencing to quantify the abundances of target proteins. Finally, the approach was shown to be capable of detecting H2B and its mono-ubiquitinated form in single cells and revealed the heterogeneities of the ubiquitination levels of H2B in single cells from different cell-cycle stages.

The data presented appear to be of high quality and robust. This alternative method has the potential to be incredibly powerful, as the approaches to interpreting the protein heterogeneity level of detail are limited. As with every proof-of-concept advance, there are many questions that remain to be addressed in this setting, but what the present study has already revealed is more than sufficient, in my opinion, for publication in Communications Biology.

Major points

1) The labeling efficiency and specificity are the basis for this work. You demonstrated that spycatcher tagging and T7 ligation efficiency are more than 90%, as estimated from the band intensity". In my opinion, the WB method failed to detect samples of broad amount dynamic range, especially with high abundance band. Do you have more information to calculate the efficiency through other approaches such as LC-MS/MS or NGS on the product and leftover bands in figure S2?

2) The researchers estimated that the detection efficiency to be $40 \pm 18\%$ and speculated the majority of loss presumably occurred during sample handling and gel electrophoresis. For the protein and its PTM isoforms ratio, the relative quantification was enough. For copy number quantification, the efficiency was required. Is the detection efficiency for general or just for H2B in this work?

Quantification results of H2B was analyzed and shown in the paper, what about the result of other proteins, such as Pre1? Could you comment on the detection efficiency in the last part of this manuscript?

3) The experiments were performed on BY4741 yeast cells. Do the researchers have any experiences about this protocol applicable to mammalian cells? Without zymolase digestion, is the spycatcher labeling efficiency and the target proteins detection efficiency higher?

4) The result in figure 2b that H2B copy numbers of the cells at the G2/M stage is roughly two-fold of the cells at the G1 stage is perfect. Were the percentages of the cells in G1 and G2/M similar with the results of flow cytometry reported before?

Minor points

5) Line 31, the reference was missed.

6) What is the meaning of rxn in figure 1d and S2a? Without any annotation.

7) Provide the full name of UMI first mentioned in line 68?

8) Check the space between works and (), such as line 77 electrophoresis(SDS-PAGE), line 108 H2B(H2Bub), line 127 Separation(DUET), figure S1 conjugate(spycatcher-20mer) and so on. Please check throughout the manuscript.

9) Line 89 : Finally, the extracted protein-oligo conjugates were made into sequencing library by "PCR"instead of"PCA"?

10) Another point, and excuse me if I missed it, is that the sequence graphic in figure 1c was identical with that in Supplemental Table? I recommended that the ligation bridge sequences were also listed in the SI table.

11) In figure S7, the distributions of H2Bub/H2B ratio analysis like in figure 2e will be valuable to understanding DUBs' potential functions in cell cycle.

12) The yeast protein name was always expressed as Pre1, not PRE1. Also check others.

Reviewer #3 (Remarks to the Author):

COMMSBIO-20-0089-T Cai

Here the authors have developed a new method, DUET, to estimate the fraction of a protein population that is post-translationally modified in a single cell. The method involves engineering a cell, in this case yeast, in which the protein of interest (POI) is fused to spycatcher, a split protein fragment that can be covalently labeled by incubating with the 13-mer spyttag peptide linked to an oligonucleotide (ODN). After labeling, small numbers of yeast cells are distributed in a 96-well plate

and labeled with spyttag conjugated to an ODN containing a forward PCR primer, and then 96 different UMI-barcode ODNs are ligated on using a T7 ligation bridge. The cells are then pooled, split again into 96 wells, and a second set of barcode-reverse primer ODNs are ligated onto the first UMI-barcode ODN. After pooling again and mixing with dummy cells with POI-linked ODNs but no primer sites, the cells are lysed and resolved on an SDS gel to separate the unmodified and modified form of the POI. In this case, the authors chose to validate their DUET method by analyzing the extent of monoubiquitylation of histone 2B, where the two forms differ by ~ 7 kDa and can readily be gel separated. The two protein bands are cut out of the gel, and the tags amplified by PCR, followed by sequencing to identify the unique cell barcodes. Analysis of the sequencing data to distinguish real barcodes from background allowed identification of 850 barcodes, corresponding to 850 single cells. The exact number of H2B and H2B-Ub molecules in a single cell was calculated from the level of the UMI sequence reads. In this analysis, the number of H2B molecules per cell was determined to be $\sim 40,000$, which is $\sim 40\%$ of the predicted H2B protein copy number, suggesting that the efficiency of labeling and recovery was about 40%. There was a bimodal distribution of H2B copy numbers, which they deduced was due to cells being either in G1 or G2 - as result of DNA replication the number of H2B molecules per cell is twice as high in G2 cells. When the number of H2B-Ub molecules per cell was used to calculate an H2B-Ub/H2B ratio, they found that the ratio was higher in G1 cells ($\sim 26\%$) than in G2 cells ($\sim 5\%$). In a yeast strain in which UBP8 and UBP9, two DUBs known to remove Ub from H2B-Ub, were deleted, they found that the overall level of H2B-Ub was increased significantly, but that most of this increase was due to elevated H2B-Ub in G1 cells.

This is a clever new approach to estimating the fraction of a protein population in a single cell that is post-translationally modified. Based on the validation data with H2B/H2B-Ub in yeast, the DUET method appears to work well. However, it is a laborious method, with many steps, and has only been validated with a best case scenario, where it is easy to separate the modified and unmodified forms of an abundant cellular protein. It would be reassuring if the authors could use DUET to estimate single cell PTM modification levels of a less abundant protein where the two isoforms are not so readily separable, e.g. if the two forms do not resolve on an SDS cell. Perhaps they could analyze the level of phosphorylation of one of the transcription factors for which they showed ODN tagging in Figure 1d.

Points: 1. While the methods section is quite extensive, it is missing some key details. The yeast cells were spheroplasted and then fixed in 1% formaldehyde. Presumably, this treatment not only fixes proteins, but also opens the cell membrane, thereby exposing the spycatcher-tagged POI to the spyttag peptide-ODN conjugate, but how the spyttag-ODN gains access to the POI-spycatcher protein needs to be explained more fully. Then, after the two steps of ODN ligation in "intact" cells, the cells were subjected to SDS-PAGE. Here it is not clear, whether the formaldehyde crosslinks were deliberately reversed, or whether this depends on boiling in sample buffer. If protein-crosslinking is not fully reversed, then this may lead to loss of the POI, since it will not migrate on the SDS gel in the right place. Finally, it is not clear how many yeast cells were distributed into each well of the 96-well plate at the two steps, and this needs to be indicated.

2. The total number of H2B protein molecules should be the number of H2B molecules plus the number of H2B-Ub molecule, but is not clear from the description on page 3 whether this is what was done.

3. The difference in H2B-Ub/H2B ratio in G1 and G2 cells is interesting, but was apparently not due to UBP9/10 DUB activity. These findings deserve further discussion, i.e. what is known about how H2B-Ub modification is regulated during the cell cycle - this will be the result of a balance between the activities of the E3 ligase and the DUB. What is the H2B E3 Ub ligase and does its level/activity change during the cell cycle, and is there a cell-cycle specific H2B-Ub DUB known that could account for the lower level of H2B-Ub in G2 cells?

4. The overall range in the number of H2B (and H2B-Ub) protein molecules in individual cells appears to be ~ 10 fold based on the data in Supplementary Figure 5d. This obviously represents a mixture of G1 and G2 cells, which might be expected to differ in H2B content ~ 2 -fold, but not more than that.

Considering that the amount of DNA per cell is very highly regulated, and that one would expect a tight correlation between the amount of DNA and H2B levels because of its role in nucleosomal packaging of DNA, this degree of variation is unexpected and merits further discussion. For instance, is this a technical issue or variation in the efficiency of spytag-ODN labeling in different cells? A wider range of H2B`Ub levels might be expected, since post-translation stoichiometry can be regulated depending on the exact physiological status of the individual cell.

Reviewers' comments:

We would like to thank all the reviewers for their positive comments and feedbacks. We have revised our manuscript according to their suggestions and comments, which greatly improve the clarity of this manuscript. We have also addressed each of the concerns point-by-point below.

Reviewer #1 (Remarks to the Author):

Zhang and co-authors develop a method to analyze post-translational modified populations of a protein at the level of single cells. It combines the use of DNA bar codes and pooling to protein gel electrophoresis and NGS for DNA sequencing. They applied the method to detect histone 2B and its ubiquitinated form. Indeed, a method to analyze PTM of proteins in single cells is lacking and would be of great interest to the community. However, the manuscript needs clarifications prior to publication. The manuscript is difficult to follow.

It would help the reader if the rationale of each step is explained. Even scheme B of Figure 1 is not clear and it is the main figure.

Thank you for your suggestion. We have improved the clarity of the manuscript by adding the rationale of each step.

We have revised the Figure 1b to make it clearer. In Figure 1b, we illustrated how single-cell barcoding is achieved by oligo tagging and combinatorial indexing. First, the cells were reacted with spycatcher-oligo to attach an oligonucleotide to the protein of interest (with spytag). Then, the protein of interest were further ligated with barcode oligos in a “split-pool” fashion to uniquely barcode every single cells.

Why two rounds of ligation are done?

In two-round “pool-split” ligation, the cells were firstly distributed into a 96-well plate. Each well contains a unique barcode oligo so that the cells were barcoded in each well. Then the cells from different wells were combined and split again into a second 96-well plate, which also contains another set of well-specific barcode oligo. During this “pool-split” process, the cells that are in the same well in the 1st round plate will be distributed into different wells in the 2nd plate to provide over 9000 unique barcodes. If we sample a small number of cells (900 cells in our experiment) then the large barcoding space allows sampled cells to be uniquely barcoded with a low collision rate of ~5%, which gave us enough statistical power for further analysis.

The use of this combinatorial indexing avoids manipulation of single cells and allows large number of single cells to be barcoded. Using 384-well plate and larger number of ligation rounds can be adapted to further increase the throughput according to experimental needs and design, as demonstrated in previous literature (Rosenberg et al., *Science* 360, 176–182 (2018))

It's not clear how it is determined which cells are in G1 and which in G2M. How copy number is measured

We have rewritten this part of manuscript to make it more clear. The protein copy number is measured by counting the unique molecular identifiers (UMIs) in single cells. Each unique UMI corresponds to one single protein.

Previous literature reported that the H2B copy number is doubled after DNA is replicated (Eriksson et al., *Genetics* 191, 7–20 (2012)). The H2B copy number in single cells measured from our method was plotted in histogram (Figure 2b). This histogram shows a bimodal distribution and the H2B copy numbers of the two peaks are roughly two-fold, which likely correspond to cells in G1 or G2/M stages.

How this analysis is related to single cell information given the accuracy (Authors detected on average of 4065 ± 1798 UMI of H2B proteins per cell with a detection efficiency to be $40 \pm 18\%$)

Based on another reviewer's suggestion, we added the H2Bub copy number to the H2B copy number, as the total H2B protein copy number detected in the cells. Using the UMIs as a measure of quantitation, we detected on average of 4568 ± 2004 (s.d.) proteins per cell.

The number of H2B proteins per cell is $45 \pm 20\%$ of the predicted H2B protein copy number (Ho et al., *Cell Syst* 6, 2015), suggesting that the overall detection efficiency (including the labeling efficiency and sample recovery efficiency) of our method was about $45 \pm 20\%$.

Finally, the methodology is limited to proteins that are tagged so it cannot be applied to native proteins. This is a big limitation to its application.

We agree with the reviewer's point. In the current demonstration of DUET, we used genetic fusion to selectively target a single protein. We believe it is possible to extend DUET to label the proteome globally without the need of genetic manipulation. This could be done by incorporation of unnatural amino acids into the proteome of cells (Dieterich et al., *Proc. Natl. Acad. Sci. U. S. A.* 103, 2006). The unnatural amino acids could be clicked to oligonucleotide and

barcoded using the same strategy demonstrated here. Then, different proteins-oligo conjugates could be separated by high dimensional gels or liquid chromatography (Washburn et al., *Nat. Biotechnol.* 19, 242–247 (2001)), followed by identification by mass spectrometry and single-cell quantification by NGS of the individual species, to enable single cell proteomics at a global scale. We have included this discussion into the main text.

Reviewer #2 (Remarks to the Author):

In this manuscript entitled “Detecting protein and post-translational modifications in single cells with iDentification and qUantification sEparaTion (DUET)”, Yandong Zhang and colleagues describe a new method they coin “DUET” that can be used to detect protein and its PTM isoforms in single cells. The authors innovatively labeled protein with oligonucleotides and combined NGS sequencing to quantify the abundances of target proteins. Finally, the approach was shown to be capable of detecting H2B and its mono-ubiquitinated form in single cells and revealed the heterogeneities of the ubiquitination levels of H2B in single cells from different cell-cycle stages.

The data presented appear to be of high quality and robust. This alternative method has the potential to be incredibly powerful, as the approaches to interpreting the protein heterogeneity level of detail are limited. As with every proof-of-concept advance, there are many questions that remain to be addressed in this setting, but what the present study has already revealed is more than sufficient, in my opinion, for publication in *Communications Biology*.

Thank you for the positive comments.

Major points

1) The labeling efficiency and specificity are the basis for this work. You demonstrated that spycatcher tagging and T7 ligation efficiency are more than 90%, as estimated from the band intensity”. In my opinion, the WB method failed to detect samples of broad amount dynamic range, especially with high abundance band. Do you have more information to calculate the efficiency through other approaches such as LC-MS/MS or NGS on the product and leftover bands in figure S2?

We agreed that WB is not accurate to estimate samples with high abundance due to its narrow dynamic range. We cannot sequence the leftover bands because it does not have the PCR handles and cannot be amplified into sequencing libraries. We also do not have access to a LC-MS at the moment. However, by comparing the product band and leftover band, WB method did provide semi-quantitative

estimates for labeling efficiency, as demonstrated in other literature (Heidebrecht et al., *J. Immunol. Methods* 345, 40–48 (2009).). These results are in agreement with previous literature (Zakeri et al., *Proceedings of the National Academy of Sciences* 109, 690–697 (2012)) showing the spyttag/spycatcher reaction and T7 ligation is very efficient.

We edited the manuscript by replacing the numeric efficiency with descriptive sentence like “The labeling efficiency is high” in the main text to be more rigorous. We also revised Figure S2 to make the leftover band more visible.

2) The researchers estimated that the detection efficiency to be $40 \pm 18\%$ and speculated the majority of loss presumably occurred during sample handling and gel electrophoresis. For the protein and its PTM isoforms ratio, the relative quantification was enough. For copy number quantification, the efficiency was required. Is the detection efficiency for general or just for H2B in this work? Quantification results of H2B was analyzed and shown in the paper, what about the result of other proteins, such as Pre1? Could you comment on the detection efficiency in the last part of this manuscript?

We did not prepare libraries from the other proteins. We plan to work on other proteins, such as Pre1, as well as global labeling of proteins with unnatural amino acids in the future. While we don't know the exact detection efficiencies for these proteins, Figure 1d shows that the labeling efficiency (spycatcher reaction and T7 ligation) for these proteins (Pre1(copies: 13312/cell), Snf1(5423/cell), Glc7(12111/cell), Ho et al., *Cell Syst* 6, 192–205.e3 (2018)) are generally as high as H2B proteins. Since the rest of our experimental protocols, including gel electrophoresis and NGS, are all the same for different proteins, we expect that the degree of sample loss for different proteins will be also similar. Therefore, we expect that detection efficiency for other proteins will be qualitatively similar as H2B shown in this work. We have added a discussion on the detection efficiency in the last part of the revised manuscript.

3) The experiments were performed on BY4741 yeast cells. Do the researchers have any experiences about this protocol applicable to mammalian cells? Without zymolase digestion, is the spycatcher labeling efficiency and the target proteins detection efficiency higher?

This is a great point. We expect the efficiency to be higher in mammalian cells. Spycatcher/spyttag labeling and serial T7 ligation has been applied in fixed mammalian cells lines with high efficiency (Pessino et al., *Chembiochem* 18, 1492–1495 (2017); Rosenberg et al., *Science* 360, 176–182 (2018)).

4) The result in figure 2b that H2B copy numbers of the cells at the G2/M stage is roughly two-fold of the cells at the G1 stage is perfect. Were the percentages of the cells in G1 and G2/M similar with the results of flow cytometry reported before?

The percentage of G2/M stage is 40% from our data. This is in agreement with flow cytometry data with 40% reported in literature in which the DNA content is stained and used to estimate cell cycle stage (Calvert et al., *Cytometry A* 73, 825–833 (2008)).

Minor points

5) Line 31, the reference was missed.

6) What is the meaning of rxn in figure 1d and S2a? Without any annotation.

7) Provide the full name of UMI first mentioned in line 68?

8) Check the space between words and (), such as line 77 electrophoresis(SDS-PAGE), line 108 H2B(H2Bub), line 127 Separation(DUET), figure S1 conjugate(spycatcher-20mer) and so on. Please check throughout the manuscript.

9) Line 89 : Finally, the extracted protein-oligo conjugates were made into sequencing library by “PCR”instead of”PCA”?

10) Another point, and excuse me if I missed it, is that the sequence graphic in figure 1c was identical with that in Supplemental Table? I recommended that the ligation bridge sequences were also listed in the SI table.

11) In figure S7, the distributions of H2Bub/H2B ratio analysis like in figure 2e will be valuable to understanding DUBs' potential functions in cell cycle.

12) The yeast protein name was always expressed as Pre1, not PRE1. Also check others.

Thanks for the reviewer's feedback. We have revised the manuscript according to these suggestions.

Reviewer 3:

COMMSBIO-20-0089-T Cai

Here the authors have developed a new method, DUET, to estimate the fraction of a protein population that is post-translationally modified in a single cell. The method involves engineering a cell, in this case yeast, in which the protein of interest (POI) is fused to spycatcher, a split protein fragment that can be covalently labeled by incubating with the 13-mer spyttag peptide linked to an oligonucleotide (ODN). After labeling, small numbers of yeast cells are distributed in a 96-well plate and labeled with spyttag conjugated to an ODN containing a forward PCR primer, and then 96 different UMI-barcode ODNs are ligated on using a T7 ligation bridge. The cells are then pooled, split again into 96 wells, and a second set of barcode-reverse primer ODNs are ligated onto the first UMI-barcode ODN. After pooling again and mixing with dummy cells with POI-linked ODNs but no primer sites, the cells are lysed and resolved on an SDS gel to

separate the unmodified and modified form of the POI. In this case, the authors chose to validate their DUET method by analyzing the extent of monoubiquitylation of histone 2B, where the two forms differ by ~ 7kDa and can readily be gel separated. The two protein bands are cut out of the gel, and the tags amplified by PCR, followed by sequencing to identify the unique cell barcodes. Analysis of the sequencing data to distinguish real bar codes from background allowed identification of 850 barcodes, corresponding to 850 single cells. The exact number of H2B and H2B-Ub molecules in a single cell was calculated from the level of the UMI sequence reads. In this analysis, the number of H2B molecules per cell was determined to be ~40,000, which is ~40% of the predicted H2B protein copy number, suggesting that the efficiency of labeling and recovery was about 40%. There was a bimodal distribution of H2B copy numbers, which they deduced was due to cells being either in G1 or G2 - as result of DNA replication the number of H2B molecules per cell is twice as high in G2 cells. When the number of H2B~Ub molecules per cell was used to calculate an H2B~Ub/H2B ratio, they found that the ratio was higher in G1 cells (~26%) than in G2 cells (~5%). In a yeast strain in which UBP8 and UBP9, two DUBs known to remove Ub from H2B~Ub, were deleted, they found that the overall level of H2B~Ub was increased significantly, but that most of this increase was due to elevated H2B~Ub in G1 cells.

This is a clever new approach to estimating the fraction of a protein population in a single cell that is post-translationally modified. Based on the validation data with H2B/H2B~Ub in yeast, the DUET method appears to work well. However, it is a laborious method, with many steps, and has only been validated with a best case scenario, where it is easy to separate the modified and unmodified forms of an abundant cellular protein. It would be reassuring if the authors could use DUET to estimate single cell PTM modification levels of a less abundant protein where the two isoforms are not so readily separable, e.g. if the two forms do not resolve on an SDS cell. Perhaps they could analyze the level of phosphorylation of one of the transcription factors for which they showed ODN tagging in Figure 1d.

Thank you for your comments. We cannot perform additional experiments at the moment and we plan on resolving phosphorylation isoform for transcription factors in a future experiment.

Points: 1. While the methods section is quite extensive, it is missing some key details. The yeast cells were spheroplasted and then fixed in 1% formaldehyde. Presumably, this treatment not only fixes proteins, but also opens the cell membrane, thereby exposing the spycatcher-tagged POI to the spytag peptide-ODN conjugate, but how the spytag-ODN gains access to the POI-spycatcher protein needs to be explained more fully. Then, after the two steps of ODN ligation in "intact" cells, the cells were subjected to SDS-PAGE. Here it is not clear, whether the formaldehyde crosslinks were deliberately reversed, or whether this depends on boiling in sample buffer. If protein-crosslinking is not fully reversed, then this may lead to loss of the POI, since it will not migrate on the SDS gel in the right place. Finally, it is not clear how many yeast cells were distributed into each well of the 96-well plate at the two steps, and this needs to be indicated.

In the revision, we have included additional detailed information in the method section and improve the clarity of the main text.

Yes, the reviewer is correct in pointing out that the 1% formaldehyde treatment not only fixed the proteins, but also permeabilized the cell membranes, so that the spycatcher-ODN can diffuse into the cells and react with POI-spytag, consistent with previous observations (Pessino et al., *Chembiochem* 18, 1492–1495 (2017)). The spheroplasting process is necessary to remove the cell wall of the yeast cells, which otherwise will block spycatcher-ODN from entering into the cells.

The formaldehyde crosslinks were reversed by the boiling in sample buffer during the standard SDS-PAGE sample preparation step. We tried several other un-crosslinking methods and found that boiling in sample buffer is sufficient to reverse the crosslinks. This is now discussed in the method section.

In the two-round “split-pool” barcoding step, $\sim 10^6$ cells were distributed into each well. Only 900 cells were sampled at the end to ensure each cell had a unique barcode. We have added this information to the supplemental methods section.

2. The total number of H2B protein molecules should be the number of H2B molecules plus the number of H2B~Ub molecule, but is not clear from the description on page 3 whether this is what was done.

Thanks for this suggestion. We added the H2Bub (on average 503 copies/cell) to H2B (on average 4065 copies/cell) for the total number of H2B protein molecules in the revised manuscript.

3. The difference in H2B~Ub/H2B ratio in G1 and G2 cells is interesting, but was apparently not due to UBP9/10 DUB activity. These findings deserve further discussion, i.e. what is known about how H2B~Ub modification is regulated during the cell cycle - this will be the result of a balance between the activities of the E3 ligase and the DUB. What is the H2B E 3 Ub ligase and does its level/activity change during the cell cycle, and is there a cell-cycle specific H2B~Ub DUB known that could account for the lower level of H2B~Ub in G2 cells?

Following this suggestion, we have included additional discussion into the manuscript. The H2B is mono-ubiquitinated by E3 ligase Bre1 and E2 ubiquitin-conjugating enzyme Rad6.

While there is no report on the activity of those enzymes during the cell cycle, it has been reported that Rad6 is phosphorylated by the Bur1/Bur2 cyclin-dependent protein kinase at its serine-120 residue, and this phosphorylation of Rad6 is required for its catalytic activity (Wood et al., *Mol. Cell* 20, 589–599 (2005)). BUR1 is an essential gene that encodes a Cdc28-related cyclin-dependent protein kinase. BUR2 encodes a divergent cyclin of the cyclin T/cyclin C family that interacts with Bur1 and is required for its protein kinase activity (Yao et al., *Mol. Cell. Biol.* 20, 7080–7087 (2000)). These findings indicate that Rad6/Bre1 and H2B-ubiquitination might be regulated in cell cycle progression. Therefore, it would be interesting to study the Bur1/Bur2 activities/levels during the cell-cycle progression. On the other hand, little appears to be known about whether there is cell-cycle specific DUB activities with Ubp8/10.

4. The overall range in the number of H2B (and H2B~Ub) protein molecules in individual cells appears to be ~10 fold based on the data in Supplementary Figure 5d. This obviously represents a mixture of G1 and G2 cells, which might be expected to differ in H2B content ~2-fold, but not more than that. Considering that the amount of DNA per cell is very highly regulated, and that one would expect a tight correlation between the amount of DNA and H2B levels because of its role in nucleosomal packaging of DNA, this degree of variation is unexpected and merits further discussion. For instance, is this a technical issue or variation in the efficiency of spyttag-ODN labeling in different cells? A wider range of H2B~Ub levels might be expected, since post-translation stoichiometry can be regulated depending on the exact physiological status of the individual cell.

Previous literature (Chong et al., *Cell* 161, 1413–1424 (2015); Ho et al., *Cell Syst* 6, 2015) reported a large variation of H2B copy number (101430 ± 63961 (s.d.)) based on measuring single-cell H2B-GFP intensity, which is likely quite quantitative. The coefficient of variation (CV) of their method is 0.63, which is similar to the CV from our experiment (CV=0.43). Thus it is likely the main source of the single cell variability in our experiment reflects accurately the expression levels of H2B rather than technical noise. Given the difficulties in controlling the level of proteins through bursty events of transcription, it is possible that there is a pool of free H2B in the nucleus that buffers the amount of histones bound on DNA and variability in H2B abundance can be functionally moderated.

REVIEWERS' COMMENTS:

Reviewer #2 (Remarks to the Author):

My comments were addressed nicely, and the manuscript has improved considerably for publication. The authors have extensively revised their manuscript including the revision and addition of novel analysis. The figure 1 has been improved and more easier to follow. Though the key point of labeling efficiency has been proved before as the authors cited the before, I strongly recommended that quantified the labeling efficiency with more accuracy method in the future study.

Reviewer #3 (Remarks to the Author):

The authors have satisfactorily addressed my concerns by adding further discussion.